# Reconstruction of Single-Bay Buddhist Architecture Based on Stylistic Comparisons in Northeast Fujian, the Core Hinterland of the Changxi River Basin—Using Gonghoulong Temple as an Example

**Yu Ding** , **Yuqing Cai and Jie Liu** *

Department of Architecture, School of Design, Shanghai Jiao Tong University, Shanghai 200240, China; dingdingkorn@163.com (Y.D.); yuqingcai@sjtu.edu.cn (Y.C.)
* Correspondence: jackliu@sjtu.edu.cn

**Abstract:** In the Changxi River Basin in eastern Fujian, a few stone elements remain and Buddhist buildings with one bay in width and three bays in depth have been preserved dating from the timespan of the Tang to the Song dynasty. These features are characterized by a regional form of early Buddhist architecture seldom seen in Chinese history. The article focuses on the reconstruction of a Song dynasty Buddhist building at the Gonghoulou Temple site in Huotong Town, Jiaocheng District, Ningde City, and aims to analyze the potential characteristics and rules of single-bay Buddhist architecture. From the styles of the remaining stone columns, the direction of the lotus carving at the column base, and the mortises of the plinth stone, a spatial arrangement is indicated that includes an open front corridor and a closed rear section. A "reconstruction" of the ruler used in the original building reveals the possibility that a local Fujian ruler was used, shorter than the standard measurement device employed elsewhere. The analysis of the frame construction indicates that this hip-gable roof-covered Buddhist hall utilizes the horizontally layered logic of multi-storied palatial-style halls. Key elements include its gentle roof slope, restraint from the practice of shortening the roof ridge, use of the traditional *chuji* method, and the interior columns use of internal longitudinal architraves secured to beam supporting brackets. This research brings to light a unique architectural type that has disappeared in the course of history and was previously unknown to the academic community. It holds significant importance and value for deepening the understanding of the history of timber frame architecture technology in Fujian.

**Keywords:** Changxi River Basin; one-bay; palatial hall; construction ruler; reconstruction

## 1. Introduction

In a previous article published by the research team of authors in Religions, a particular style of Buddhist temples popular in the region from the 9th to 12th centuries AD was revealed through analysis of architectural and stone column remains in the Changxi River Basin. In this style, the eave columns usually used two rows of pumpkin-shaped circular stone columns 瓜楞石柱 to form a longitudinal rectangular space of "one bay in width and multiple bays in depth" with no columns in the interior and adopting a hip-gable roof 歇山顶. The formation and later evolution of this one-bay Buddhist temple have been discussed (Liu et al. 2021). According to current incomplete statistics, there are seven examples of stone column remains of single-bay Buddhist architecture with the above characteristics in the Changxi River Basin and its bordering areas. In addition, there are four suspected cases that may reflect single-bay rules (Table 1). In this paper, research and reconstruction will be conducted on a Song Dynasty Buddhist temple site in Huotong Town 霍童镇, Jiaocheng District 蕉城区, Ningde City 宁德市. The purpose of the reconstruction

is through present evidence and clues to deeply explore the possible formal characteristics and rules of this one-bay Buddhist temple. On this basis, the distinctive construction system in the Changxi River Basin, including the ground plan pattern, construction rulers 营造尺 utilized, and timber frame construction techniques, will be discussed.

**Table 1.** Remains of single-bay buddhist architecture in the Changxi River Basin and its border areas.

| | No. | Name | Location | Construction Time | Source | Stone Column Remains |
|---|---|---|---|---|---|---|
| **Stone Column Remains of Single-bay Buddhist Architecture** | 1 | Chanji Temple 禅寂寺 | Huotong Town 霍童镇, Jiaocheng District 蕉城区 | The 5th year of Xiantong 咸通五年 (864) | *Ningde County Annals* of Jiajing period 嘉靖 《宁德县志》 | 2 × 3 ○ |
| | 2 | Guoxing Temple 国兴寺 | Taimushan Town 太姥山镇, Fuding City 福鼎市 | The 4th year of Qianfu 乾符四年 (877) | *Taimu Mountain Annals* of Wanli period 万历 《太姥山志》 | 2 × 4 □ |
| | 3 | Chanjini Temple 禅寂尼寺 | Gantang Town 甘棠镇, Fu'an City 福安市 | Qianhua period 乾化年间 (911–913) | | 2 × 3 □ |
| | 4 | Xingqing Temple 兴庆寺 | Xitan Town 溪潭镇, Fu'an City 福安市 | Kaibao period 开宝年间 (968–976) | | 2 × 3 ○ |
| | 5 | Bao'en Temple 报恩寺 | Xibing Town 溪柄镇, Fu'an City 福安市 | Yuanfu period 元符年间 (1098–1100) | *Fu'an County Annals* of Wanli period 万历 《福安县志》 | 2 × 2 ○ |
| | 6 | Suoquan Temple 锁泉寺 | Xiaoyang Town 晓阳镇, Fu'an City 福安市 | Yuanfu period 元符年间 (1098–1100) | | 2 × 2 ○ |
| | 7 | Sanbao Temple 三宝寺 | Chengnan Sub-district 城南街道, Fu'an City 福安市 | The 5th year of Chunyou 淳祐五年 (1245) | | 2 × 4 ○ |
| **Cases Reflect Single-bay Clues** | 1 | Jinbei Temple 金郶寺 | Jinhan Township 金涵乡, Jiaocheng District 蕉城区 | The 8th year of Dazhong 大中八年 (854) | *Ningde County Annals* of Jiajing period 嘉靖 《宁德县志》 | 2 × 3 ○ |
| | 2 | Shifeng Temple 狮峰寺 | Xibing Town 溪柄镇, Fu'an City 福安市 | The 4th year of Qianfu 乾符四年 (877) | *Fu'an County Annals* of Wanli period 万历 《福安县志》 | 2 × 4 □ |
| | 3 | Jinfeng Temple 金峰寺 | Yangzhong Town 洋中镇, Jiaocheng District 蕉城区 | The 5th year of Chunhua 淳化五年 (994) | *Ningde County Annals* of Jiajing period 嘉靖 《宁德县志》 | 2 × 2 ○ |
| | 4 | Xingyun Temple 兴云寺 | Xibing Town 溪柄镇, Fu'an City 福安市 | Yuanfu period 元符年间 (1098–1100) | *Fu'an County Annals* of Wanli period 万历 《福安县志》 | 2 × 2 ○ |

The figures in the last column refer to the number of stone columns in the direction of longitude and latitude. □ represents a square stone column, while ○ represents a pumpkin-shaped circular stone column.

## 1.1. Purpose of the Conceptual Reconstruction and Approach

The architectural reconstruction examined the remains of a Song Dynasty Buddhist temple (name of the cultural relics protection unit: Site of Gonghoulong Temple 宫后垄寺遗址) in Huotong Town, Jiaocheng District, Ningde City, as an example. The reconstruction is based on the relics of the local Buddhist temples of the Song Dynasty and the remains of stone columns, as well as later wooden architecture that evolved from the one-bay form. The first step of the reconstruction research is to summarize the characteristics of the site and model the construction ruler originally used through on-site measurement. Secondly, the characteristics of the spatial form are analyzed by interpreting the information of the remains on stone columns. Finally, the reconstruction design can be carried out through frames based on the early wooden architecture in the Changxi River Basin and the East Fujian area.

It should be noted that the local remaining wooden architecture, which is the basis for reconstruction, was frequently renovated or altered in past dynasties. Wooden com-

ponents replaced original Song Dynasty materials, and some construction does not reflect Song Dynasty styles, thus creating difficulties for study. As a result, the research focuses on the form of the frame, an architectural characteristic that is least likely to be disturbed, and traces its historical evolution in order to understand the characteristics and rules of corresponding forms. In addition, for items such as small wooden doors, windows, and roof tiles for which evidence is seldom available, we could only concentrate on stylistic reconstruction according to the Song-style historical records. While history cannot be repeated, its underlying patterns can be traced. The proposed reconstruction plan, based on the above principles, represents an attempt in the form of personal interpretation. Its significance lies in unearthing a unique architectural typology that disappeared in the course of history and was previously unknown to the academic community, thus providing a more comprehensive understanding of the history of wooden construction techniques.

*1.2. Overview of the Site History*

The Gonghoulong Temple Site is located in Xiaoshi Village 小石村, Huotong Town 霍童镇, and is currently listed as a cultural relics protection unit in Jiaocheng District 蕉城区, Ningde City 宁德市. Huotong has been a sacred Buddhist site on the southeastern coast since Wuyue Kingdom 吴越国 (907–978). The Zhiti Mountain 支提山 in the territory is known by the praise of "Without reaching Zhiti Mountain, one's journey as a monk would be in vain 不到支提枉为僧". During the Ming Dynasty, it was described by Emperor Yongle as "the best mountain in the world 天下第一山", which shows the prosperity of Buddhism in this area.

The historical information of the temple where the site is located is very limited, and there is no clear literature providing the exact name or historical timeline of the temple. There are three remaining inscriptions on stone tablets, i.e., "住持沙门崇晶誌 (recorded by Sramana Chongyan)", "住山比丘慧舟重造 (reconstructed by Bhikkhu Huizhou)", and "宋元豐四年辛酉歳十月日誌 (recorded on the day of October in the 4th year of Yuanfeng period of the Song Dynasty)". The claim that the temple was established in the 4th year of Yuanfeng (1081), as mentioned in *An Atlas of Chinese Cultural Relics: Fujian Volume* 中国文物地图集福建分册, is based on the stone inscription (National Cultural Heritage Administration 2007, p. 734). According to the *Ningde County Annals* 宁德县志 of the Jiajing period of the Ming Dynasty, "禅寂寺，在十三都，唐咸通五年建 (Chanji Temple, built in the 5th year of Xiantong period of the Tang Dynasty, is located in the 13th County.)". Xiaoshiling, where the site is located, was within the jurisdiction of the 13th County during the Ming Dynasty, which may have some connection with the temple.

There are currently eight stone columns, a Sumeru stone podium (symbolic of Buddhist Mount Sumeru) 须弥座, as well as stone grooves, stone tablets, and remaining stone architectural elements at the site. The features and carving techniques of the stone columns and Sumeru platform are consistent with other Buddhist temple remains from the Song Dynasty in the Changxi River Basin (Liu 2018). They are well preserved and located in their original positions without changes throughout the ages, indicating the feasibility of architectural reconstruction.

## 2. Analysis of Clues for the Main Hall Reconstruction

*2.1. Site Characteristics and Ruling Principles*

The eight stone columns remaining in the main hall are positioned from the northwest to the southeast, arranged in two columns with four rows, forming a layout of one-bay wide and three-bay deep. The concave-shaped Sumeru platform is located in the middle of the last two bays in a depth-wide direction (Figure 1).

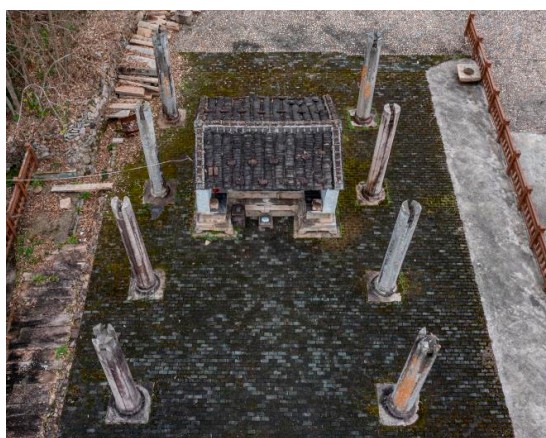

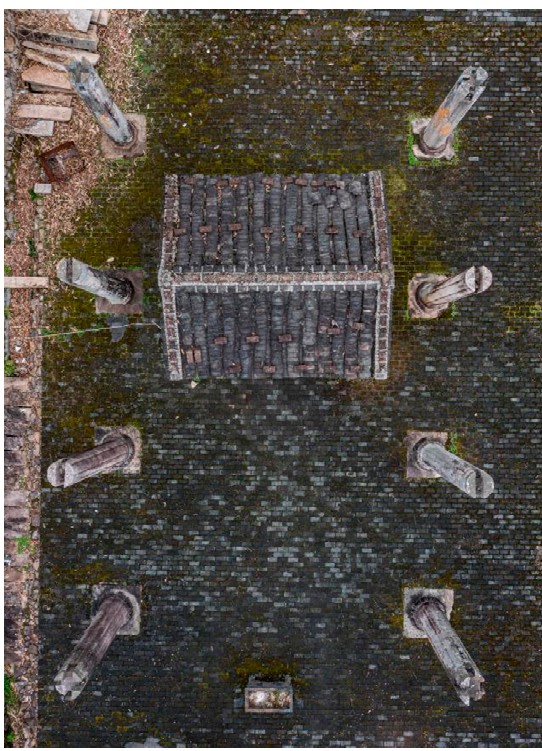

**Figure 1.** Aerial photo of the site. Photography by Jiangling Liu.

Through on-site measurement, the plan composition of the column network of the main hall is as follows:

Width: 7027 mm

Depth: 3547 + 3546 + 3541 = 10,634 mm

We used the Fujian local ruler 闽乡尺 measuring about 270 mm to serve as the lower limit per unit, and a Qing Dynasty official ruler 清官尺 measuring 320 mm as the upper limit of unit measurement (Li 2014). With incremental lengths of 1 mm, the units used by the main hall construction ruler were calculated according to the design principle of integer lengths between columns 整数尺柱间制.[1]

When the ruler length was tested at 294 mm, the width and depth of the main hall were close to whole numbers, as seen in Table 2.

**Table 2.** Reconstruction of the main hall construction ruler of Gonghoulong Temple Site (1 *chi* 尺 = 294 mm).

|  | Total Width | The First Bay of the Depth | The Second Bay of the Depth | The Third Bay of the Depth | Total Depth |
|---|---|---|---|---|---|
| mm | 7027 | 3547 | 3546 | 3541 | 10,634 |
| *chi* | 23.901 | 12.065 | 12.044 | 12.010 | 36.170 |
| Round up | 24 | 12 | 12 | 12 | 36 |
| Rate of coincidence | 99.59% | 99.46% | 99.49% | 99.63% | 99.53% |

According to the above table, the main hall construction ruler length is 294 mm, shorter than the Song official ruler 宋官尺 (about 310 mm), which should be because of the use of a Fujian local ruler, as detailed in the analysis below. In this way, the plan composition of the column network can be converted as follows:

Width: 24 *chi*

Depth: 12 + 12 + 12 = 36 *chi*

At this time, the depth of each bay is divided into two step frames 二步架, thus making the total depth six step frames 六步架. Each frame length 平长 is 36/6 = 6 *chi*, which is precisely in accordance with the rafter specifications 用椽之制 outlined in Volume Five of the Song Dynasty's architectural standard *Yingzao Fashi* 营造法式 (Wang 2023, p. 520).

### 2.2. Stone Columns Features and Spatial Form of the Main Hall

Each stone column at the site is about 3520 mm high and is composed of two parts: The column base 柱础 and the column shaft 柱身 (Figure 2). The plan of the column stone is about 1080 mm square, with a central protruding basin stone 覆盆 that is about 100 mm high. It has a mortise for installing a plinth stone 石地栿, and the surface of the protruding basin is also carved with lotus patterns, which is similar to the lotus style "铺地莲华" described in *Yingzao Fashi* (Figure 3). The column shaft can be divided into the pedestal 礩 and the body. The pedestal is approximately 140 mm high with a diameter of about 550 mm. The surface of the body is carved like a pumpkin shape. The middle part of the body has a diameter of about 496 mm, tapering to approximately 455 mm at the top. There is a mortise at the top of the column for installing longitudinal architraves (*lan'e* 阑额), measuring approximately 370 mm in height and 140 mm in width. Four corner columns have cross-shaped mortises, while the other four columns have line-shaped mortises, indicating the longitudinal architraves were installed in a circle at the column tops.

The carving pattern of lotus petals on the protruding basin stone and the orientation of the mortises are closely related to the spatial form of the main hall.

These eight stone columns have a special carving style on the protruding basin stone. While two front eave columns are entirely carved with lotuses, the other six columns combine lotus carving with plain surfaces. As shown in Figure 4, three types of lotus carvings are used for covered basin stones: Full lotus, three-quarter lotus, and half lotus. Among them, the two front eave columns (W1, E1) have a full lotus without mortises for plinth stone. The west and east front columns (W2, E2) and the two rear eave columns (W4, E4) have a three-quarter lotus with mortises opening at a 90-degree angle between the lotus and the plain surface. The west and east rear columns (W3, E3) have a half lotus with mortises positioned opposite each other.

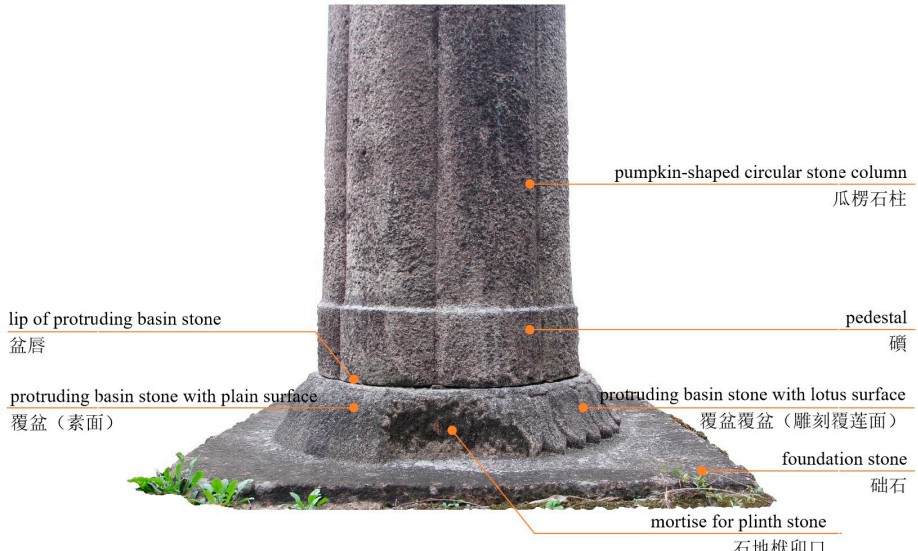

**Figure 2.** The base and foot of the remaining stone column (north elevation of W2). Drawing by authors.

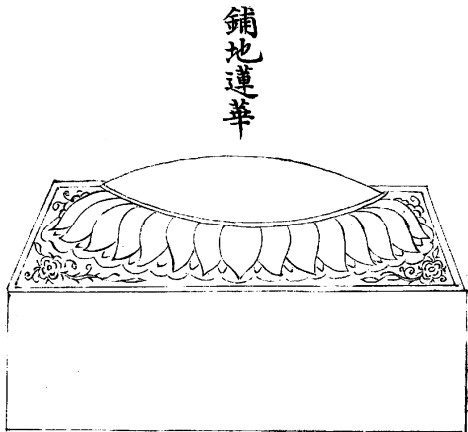

**Figure 3.** The pattern of lotus flower column base paving "铺地莲华" recorded in Volume Twenty-Nine of *Yingzao Fashi*.

The range of lotus carving and orientation of the mortises for plinth stone have a clear directionality towards the enclosing form of the main hall space. Six stone columns enclose a hall space with a width of one bay and a depth of two bays, while the two front eave columns (W1, E1) without plinth mortises indicate that the main hall features an open front corridor with a depth of one bay. The range of lotus carving refers to the differences between the interior and exterior of the main hall, i.e., the exterior of the column base is decorated with lotus carvings, while the interior of the hall remains unadorned.

In fact, it was common during the Tang and Song dynasties, particularly popular in the regions of Jiangnan and Fujian, to use the form of a Buddhist temple with an open front corridor and show the differences between interior and exterior spaces through different architectural decorations. Through the investigation of historical traces of the main hall of Baoguo Temple in Ningbo 宁波保国寺大殿, Zhang Shiqing discovered that the phenomenon of different forms of pumpkin-shaped circular columns also points to the original spatial layout of the hall with an open-fronted corridor (S. Zhang 2012a). The architectural design and intention reflected in the Gonghoulong Temple align completely with the main hall in Baoguo Temple, only with differences in the means of expression on the column bases or shafts.

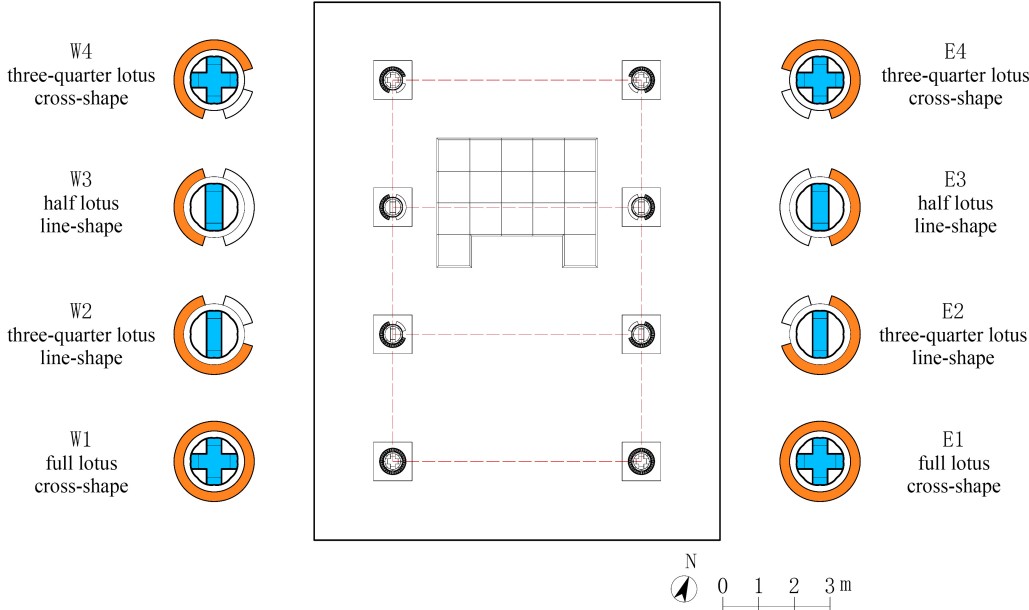

**Figure 4.** Diagram of the distribution of different styles of stone columns. Drawing by authors.

## 3. Analysis of the Characteristics of the Timber Frame Construction

There are no remaining single-bay Buddhist architecture remains in the Changxi River Basin. Due to this situation, the Buddhist architecture of the same period in eastern Fujian and neighboring areas are important references for the reconstruction of the timber frame construction of the main hall, such as the main hall of Hualin Temple in Fuzhou 福州华林寺大殿, the main hall of Chen Taiwei Palace in Luoyuan 罗源陈太尉宫正殿, architectural complex of Ganlu Temple in Taining 泰宁甘露庵建筑群, etc. Architecture rebuilt in later generations using stone columns from the Song Dynasty also has value as references, such as the main hall of Sanbao Temple in Fu'an 福安三宝寺大殿, the main hall of Shifeng Temple in Fu'an 福安狮峰寺大殿, and the Zushi hall of Mingshanshi in Yongtai 永泰名山室祖师殿.

### 3.1. Speculation on the Dimension of the Cai Module

The official construction book of the Song dynasty, *Yingzao Fashi*, recorded the design principle of adopting the *cai-fen* system 材分制 to control the size and proportionality of the building components on different scales. It clearly defined the cross-sectional dimensions of the eight grades of *cai* 材, with the height of *cai* determined as 15 *fen* 分, the thickness of *cai* 10 *fen*, and the height of *zhi* 栔6 *fen* (Guo 1999, pp. 75–82; Feng 2012, p. 33). The *cai-fen* system in *Yingzao Fashi* actually represents the ancient Chinese modular system. It specifies the scale of buildings by employing eight grades of *cai* and then controls the dimensions of various components using the basic unit of *cai-fen*. This creates concise proportional relationships, transforming complex architectural design and component processing into standardized and systematized practices. Therefore, converting actual dimensions into *cai-fen* units and determining the grade of *cai* to control the scale of the building components takes priority in the reconstruction of timber frame construction.

Focusing on the timber frame architecture from the Five Dynasties to the Southern Song Dynasty in Fujian, the height-to-depth ratio of the *cai* and the height ratio of *cai* and *zhi* both average around 2:1 (Table 3). *Yingzao Fashi* records the height of the *cai* module as 15 *fen* 分, its depth as 10 *fen*, and the height of the *zhi* as 6 *fen*. This produces a height-depth ratio of 3:2 and the height ratio of *cai* and *zhi* of 5:3. Consequently, the early timber frame architecture in Fujian exhibits a significant deviation from the measurements in the *Yingzao Fashi*, i.e., the cross-section of *cai* is more vertically elongated, and the height difference between bracket arm 栱 and tie beam 枋 is smaller, which shows distinct regional

characteristics. Based on the above, the reconstruction height-thickness ratio of *cai* and the height ratio of *cai* and *zhi* are both determined to be 2:1.

**Table 3.** The dimensions of *cai* and *zhi* of early timber frame buildings in Fujian.

| | Height of *Cai* (mm) | Thickness of *Cai* (mm) | Height-Thickness Ratio of *Cai* | Height of *Zhi* (mm) | Height Ratio of *Cai* to *Zhi* | Data Source |
|---|---|---|---|---|---|---|
| The main hall of Hualin Temple | 307.5 | 163.6 | 1.88:1 | 140.8 | 2.18:1 | (Sun 2012, p. 75) |
| The Shen Pavilion of Ganlu Temple | 185 | 85 | 2.18:1 | 100 | 1.85:1 | (B. Zhang 1982, pp. 118–43) |
| The main hall of Chen Taiwei Palace | 190 | 90 | 2.11:1 | 85 | 2.24:1 | (Ruan 2016, pp. 230–34) |

The determination of the actual dimensions of the *cai* involves two sets of data that serve as crucial references. One is the dimensions of the mortises in the column tops used to install the longitudinal architrave, and the other is the diameter of the columns. By converting the corresponding component dimensions into *cai-fen* values according to *Yingzao Fashi*, the reasonableness of the reconstructed dimensions of the *cai* module values can be assessed.[2]

When the calculation is based on the dimensions of mortises of the longitudinal architrave:

By converting the recorded dimensions of the longitudinal architrave in *Yingzao Fashi* into *cai-fen* values, the height is 30 *fen* and the depth is 20 *fen* (Pan and He 2005, pp. 71–72). The measured height and width of the mortise of the longitudinal architrave at the top of the columns are approximately 370 mm and 140 mm, resulting in a height-to-width ratio of about 2.64. It is slightly larger than the proportions in *Yingzao Fashi*. If the dimensional height of the longitudinal architrave, 370 mm, is converted to 30 *fen*, the dimensions of a *cai* would be 185 mm × 92.5 mm or 247 mm × 123 mm, which resemble the dimensions of the sixth-grade *cai* 六等材 and second-grade *cai* 二等材 in *Yingzao Fashi*. The sixth-grade *cai* is on the smaller size, and is typically used for pavilions or small halls, with the Shen Pavilion of Ganlu Temple 甘露庵蜃阁 forming the only matching example. The second-grade *cai*, suitable for five to seven-bay halls with double-eaved roofs, appear oversized (Wang 2023, pp. 356–57). If the width of the longitudinal architrave, 140 mm, corresponds to 20 *fen*, the resulting dimensions are even smaller than the sixth-grade *cai*. Hence, it can be ruled out. Therefore, using the dimensions of the mortises of the longitudinal architrave as a reference for the reconstruction is unreasonable.

When the calculation is based on the diameter of the columns:

It is recorded in *Yingzao Fashi* that column diameters for *diange* 殿阁 range from 42 *fen* to 45 *fen* and for halls of *tingtang* 厅堂 is 36 *fen* (Pan and He 2005, pp. 66–67). It can be presumed that the main hall should be a *diange*-style structure 殿阁式构架 with stone columns of the same height and large diameters. Assuming a central diameter of 496 mm corresponds to 45 *fen*, the dimensions of *cai* are 165 mm × 83 mm or 220 mm × 110 mm. The former is slightly undersized and thus unsuitable. The latter corresponds to the dimension of the fourth-grade *cai* 四等材 in *Yingzao Fashi* and is suitable for a three-bay palatial-style hall (Wang 2023, pp. 356–57). The main hall of Baoguo temple has a similar scale to the site and also uses fourth-grade *cai* (S. Zhang 2012b, pp. 111–15). Therefore, the fourth-grade *cai* is more reasonable.

In conclusion, the dimensions used for the reconstruction of the main hall are 220 mm × 110 mm, approximate to the fourth-grade *cai* in *Yingzao Fashi*.

### 3.2. Characteristics of the Timber Frame

### 3.2.1. Horizontal Layered Logic of the Palatial Hall

The patterns recorded in Volume Thirty-One of *Yingzao Fashi* ([Wang 2023](#), pp. 1872–93) point to two types of timber frame structures prevalent during the Song Dynasty, namely *diange*-style structure 殿阁式构架 and *tingtang*-style structure 厅堂式构架.[3] Since Liang Sicheng's annotations on *Yingzao Fashi*, research on the characteristics and differences of these two types of structures has been one of the central topics in the discussion of ancient Chinese architectural history. Scholars such as Chen Mingda ([Chen 1981](#), pp. 107–17), Fu Xinian ([Fu 2008](#), pp. 453–59), and Pan Guxi ([Pan and He 2005](#), pp. 23–29) have extensively discussed this matter. Zhang Shiqing ([S. Zhang 2012a](#), p. 119), combining previous research, provides three key indicators for distinguishing between these two types of structures: Firstly, in terms of structural logic, *diange*-style structure and *tingtang*-style structure are characterized by horizontal layering and vertical framing connections, respectively. Secondly, in terms of the relationship between cross beams 梁栿 and *puzuo* (i.e., bracket sets) 铺作, *diange*-style structure uses two sets of corss beams: the upper layer of rough (i.e., unfinished) structural framework (concealed by a ceiling) 草栿梁架 is pressed on the bracket sets, and the lower layer of exposed beam framework (not concealed by a ceiling) 明栿梁架 is hauled into bracket sets. On the other hand, a *tingtang*-style structure only uses an exposed beam framework. Thirdly, in terms of the connection between beams and columns, in *diange*-style structure, columns and beams are indirectly connected through bracket sets, while in *tingtang*-style structure, internal columns are raised, beam tails are inserted into the columns, and beams and columns are directly tied together. These three indicators are essential for distinguishing between typical *diange*-style structure and *tingtang*-style structure.

Based on the above indicators, it can be determined that for Buddhist architecture with eave columns of equal heights and without additional interior columns, adopting a *diange*-style structure with a horizontal layered logic is the most reasonable choice. The early timber frame architecture in Fujian, such as the front corridor of the main hall of Hualin Temple and the columns of equal height in the main hall of Chen Taiwei Palace, especially the common feature of beams hauling in into bracket sets and supporting the ceiling, provides clues to the possible existence of *diange*-style structure in Fujian (Figure 5) ([Xie 2016](#), pp. 23–24).

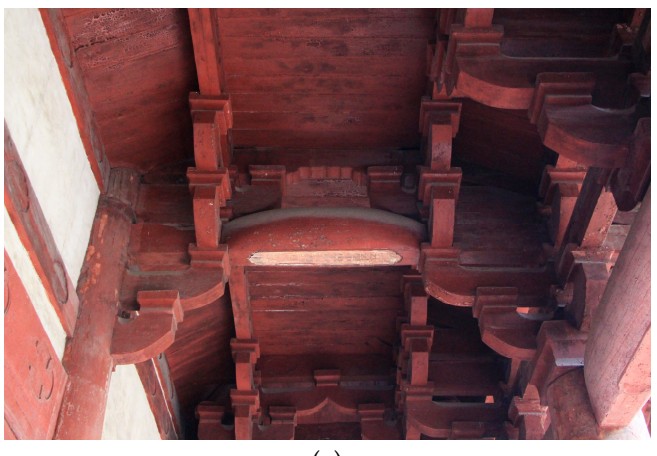
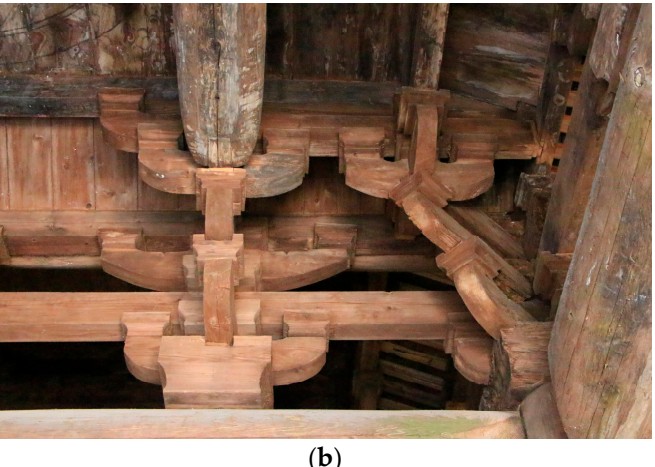

(**a**)    (**b**)

**Figure 5.** Clues for the form of palatial hall frames in eastern Fujian: beams intertwine with bracket sets to support the ceiling: (**a**) The front corridor of the main hall in Hualin temple; (**b**) the Song structure of Chen Taiwei Palace. Photography by authors.

Buddhist architecture one bay in width and multiple bays in depth in eastern Fujian display a clear and distinct horizontal layered logic. Above the longitudinal architraves,

the frame construction consists of a multi-layered bracket set with exposed beam frameworks and rough structural frameworks. The bracket layer formed by placing uniformly sized bracket sets above the longitudinal architrave, which ties together the cross beams, and supports horizontal components such as different types of tie beams 枋. The visible beams below the ceiling are finely crafted and decorated, while the beams above the ceiling support upper-level purlins and rafters with an unfinished structural framework. The rough structural frame uses the *chuandou* (column-and-tie-beam structure) 穿斗式构架, as seen in structures like Mingshan Hall in Yongtai, Sanbao Temple in Fu'an, and the main hall of the Shifeng Temple. These structures employ a simplified and lightweight roof truss through the method of connecting and linking columns with tie beams between rough structural frameworks and support the purlins directly by columns.

The rough structural frame of *chuandou* is a common form used in *diange*-style structures in the south. The earliest remaining examples can be traced back to the Southern Song Dynasty, observed in the Sanqing Hall of Xuanmiao Temple in Suzhou 苏州玄妙观三清殿. It became prevalent in the south during the Ming and Qing periods, as seen in the main hall of Shisi Temple in Jingning 景宁时思寺大殿, the Dacheng Hall of Confucius Temple in Zhangpu 漳浦县文庙大成殿, the Dacheng Hall of Confucius Temple in Quanzhou 泉州府文庙大成殿, and the main hall of the Kaiyuan Temple in Quanzhou 泉州开元寺大殿, etc. Even in cases of local official architecture, such as the main hall of the Bao'en Temple in Pingwu 平武报恩寺大殿, and the Yuantong Hall of Puji Temple on Mount Putuo 普陀山普济寺圆通殿, adopt this method, illustrating its longstanding regional significance in southern areas (Long 2013, pp. 33–36).

On the other hand, the front corridors of both the main hall of Hualin Temple and Baoguo Temple are decorated with ceilings that exhibit the high-grade characteristics of *diange*-style structure. However, the interior spaces containing Buddha statues still utilize the typical *tingtang*-style structure. Regarding this, is it possible for the restored main hall to adopt this form? Since there are no columns inside the hall, for the hip-gable roof commonly used with such frameworks in this region, the main beams of the two gables need to be supported internally. Given the depth of the main hall, the span of the main beams reaches nearly ten meters, posing significant challenges in terms of materials and construction. However, even if feasible, the installation of continuous longitudinal beams will make the framework of the main hall similar to the practice of the main hall of the Nanchan Temple in Wutai Mountain. On this basis, non-grounded columns varying in height according to the slope of the roof can be installed, but this practice does not seem to have been observed in local actual cases. At the same time, installing continuous longitudinal beams would eliminate the distinction between the front corridor and the Buddhist statue space, erasing the original intention of differentiated spatial design. However, separating the rough structural framework by installing ceilings can both reduce the roof load and present a more integrated interior space. For the reconstruction plan, this is a logical choice.

### 3.2.2. Type and Slope of the Roof

In the Changxi River Basin and even in Fujian, early Buddhist architecture is speculated to have had hip-gable roofs. Summarily, there are two distinctive features of the hip-gable roof technology in this region, i.e., the minimal or non-existent use of the *shoushan* 收山 technique for shortening the roof-ridge, and the adoption of *chuji* 出际 technique for extending the rafters beyond the edge of the gable wall. The latter approach involves supporting the gable eave rafters by either a system of gable beams or through a *chuandou* column-and-tie-beam structure that serves to extend the starting point of *chuji*.[4]

Having minimal or no *shoushan* is the most particular characteristic of the hip-gable roofs in the Changxi River Basin and even in Fujian (Lin 2014, pp. 124–25). In early wooden structures such as the main hall of Hualin Temple and the Shen Pavilion of Ganlu Temple, there is noticeable use of *shoushan*. However, there is no *shoushan* in the Chen Taiwei Palace

in Luoyuan, the upper hall, the Nan'an Hall, or the Guanyinge of Ganlu Temple. Additionally, there is minimal *shoushan* in the Zushi Hall in Mingshan Temple in Yongtai.

In the main hall of the Chen Taiwei Palace, specifically its section from the Song Dynasty, two additional frames were positioned in the range of the original single bay, and the space between intercolumnar bracket sets 补间铺作 near the two gables. Then frames on two gables were set to extend the starting point of the *chuji*. The original location of the bargeboard was outside the axis of the outer row of columns. The roof truss was enlarged through subsequent renovations, which resulted in the original Song's gable frame and bargeboard being concealed within the massive roof truss. The rough structural framework over the middle bracket sets supports two eave rafters, simultaneously extending the bargeboard beyond the original Song structure, forming the roof and the massive bargeboard to cover the entire frame (Figure 6).

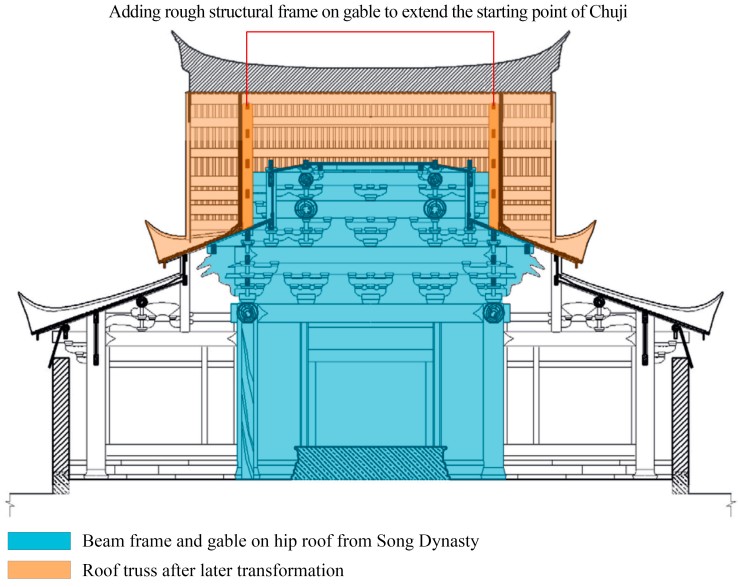

Adding rough structural frame on gable to extend the starting point of Chuji

■ Beam frame and gable on hip roof from Song Dynasty
■ Roof truss after later transformation

**Figure 6.** The frame of hip-gable roof in the main hall of the Chen Taiwei Palace. Drawing by authors and the base plan cited from (Ruan 2016, p. 234).

The rough structural framework of *chuandou* with a hip-gable roof and *chuji* in Buddhist architecture in the Changxi River Basin share a similar logic of transformation as seen in the main hall of the Chen Taiwei Palace. Examples include the main halls of Sanbao Temple and Shifeng Temple in Fu'an, both of which have *chuji* at the position of the edges of the eave columns supported by *chuandou* structures. Consequently, the tails of the rafters from the two gables are connected at the tie beams of the *chuandou* structure. Due to limited extension, the lower part of the gable only has a frame of one-rafter but maintains the traditional approach of *chuji*.

Regarding the slope of the roof, among early wooden structures in Fujian and a few wooden Buddhist architectures in the Changxi River Basin, the ratio between the rise in the eave and the distance between the center of the front and rear eaves purlins 前后橑檐枋心距 is below 1:4.0 (Table 4). Although there seems to have been an increase during the Ming and Qing periods, this ratio is still significantly different from the 1:3 specified in Volume Five of *Yingzao Fashi* (Wang 2023, p. 537). Therefore, the roof slope tends to be more flat with a minimal depression, forming a gently curved roofline. This retains the distinct regional construction characteristics of Fujian.

**Table 4.** The example data of roof slope of wooden architectures in Fujian.

| | Height of Eave H (mm) | Distance between the Center of the Front and Rear Eaves Purlins D (mm) | H/D | Data Source |
|---|---|---|---|---|
| The main hall of Hualin Temple | 4578 | 18,727 | 1:4.1 | (Sun 2012, p. 86) |
| The Shen Pavilion of Ganlu Temple | — | — | 1:4.4 | (B. Zhang 1982, pp. 118–43) |
| The Upper Hall of Ganlu Temple | — | — | 1:4.4 | |
| The main hall of the Chen Taiwei Palace (speculation of Song Dynasty) | 3115 | 13,109 | 1:4.2 | (Ruan 2016, pp. 230–4) |
| The main hall of the Chen Taiwei Palace (current situation) | 4078 | 16,688 | 1:4.1 | |
| The main hall of Sanbao Temple in Fu'an | 3107 | 12,544 | 1:4.0 | Measuring on site |
| The main hall of Shifeng Temple in Fu'an | 3610 | 14,320 | 1:4.0 | *Protection Plan of Shifeng Temple* |

### 3.3. The Configuration and Type of Bracket Sets

The configuration of the two intercolumnar bracket sets in the central bay is similar both to that recorded in *Yingzao Fashi* (Pan and He 2005, pp. 81–83), as well as to the front eave of the main hall of Hualin Temple in Fuzhou and the upper hall of the Ganlu Temple in Taining. Additionally, the Zushi Hall of Mingshan Temple in Yongtai exemplifies the architectural style of Song Dynasty structures in Fujian. In these cases, there are two intercolumnar bracket sets in the front and rear eaves. Based on the preceding analysis and the results of the main hall's reconstruction, the main hall has a central bay width of 24 *chi* and a depth of three bays, each measuring 12 *chi*. Thus, there should have been two intercolumnar bracket sets in the central bay in the longitudinal direction, and one intercolumnar bracket set in each bay in the transverse direction.

Regarding the bracket sets type, the main hall of Hualin Temple in Fuzhou and the Sanqing Hall of Yuanmiao Temple in Putian exhibit the highest level, utilizing the seven-tiered bracket sets 七铺作. The second highest level can be seen in the main hall of the Chen Taiwei Palace in Luoyuan, employing a six-level-tiered bracket set 六铺作. While the use of *xia'ang* 下昂 (downward cantilever) should be considered in conjunction with the building grade, materials, and the era in which it was constructed. High-level grade buildings with larger-sized *cai* from the Five Dynasties and Northern Song Dynasty adopted real *xia'ang*, while the main hall of the Chen Taiwei Palace in the Southern Song Dynasty as a smaller-scale regional temple utilized *cha'ang* 插昂 (S. Zhang 1999). As for the small-scale temples such as the architectural complex of Ganlu Temple in Taining and Zushi Hall of Mingshan Temple in Yongtai, the use of *ang* is no longer observed. In the cases where the *xia'ang* extension of brackets was used, the "mocking head" *shatou* 耍头 were processed in the form of the *xia'ang*.

It is noteworthy that this approach seems to have elevated the architectural rank of the structure. In these cases, *linggong* (shorter arms) 令棋 are used at the location of the upper *ang* to support tie beams, thus lowering the height of the eaves extension. The simple bracket arm and tie beams overlap to form a wall supporting bracket arm 扶壁棋. The large size of the *ludou* 栌斗 and the practice of using three-lobed curves at the apex of *ang* are all distinctive regional characteristics found in eastern Fujian and even in Fujian as a whole.

Taking into account the aforementioned hierarchical relationships and the scale of the bracket sets in the main hall, it is proposed to adapt a five-tiered bracket set 五铺作.

### 3.4. Nei'e (Internal Longitudinal Architraves) and Connected Beam Bearing Bracket Technology

For the one-bay main hall with hip-gable roof, the special feature lies in the form and position of the beam frame. The reconstruction shows that there are no columns inside the main hall, so except for the front and rear eaves, the remaining four columns in the east–west direction must have internal longitudinal architraves to support the upper ceiling, and the rough structural framework is supported by the intercolumnar bracket sets on the architrave. However, based on the actual situation, these four columns lack east–west mortises for installing architraves. Thus, it is inferred that architraves are actually connected with bracket sets on top of the columns, which extend outward to serve as bracket arms supporting the eaves, as can be seen in the central column in the Zushi Hall of Mingshan Temple in Yongtai. The *ludou* above the two central columns supports a moon-shaped beam 月梁 that spans the width. Two bracket sets are on the longitudinal columns, and the forward and rear bracket sets center each have a *huagong* bracket arm 华栱. The top of the brackets then supports the cross and tie beams (Figure 7).

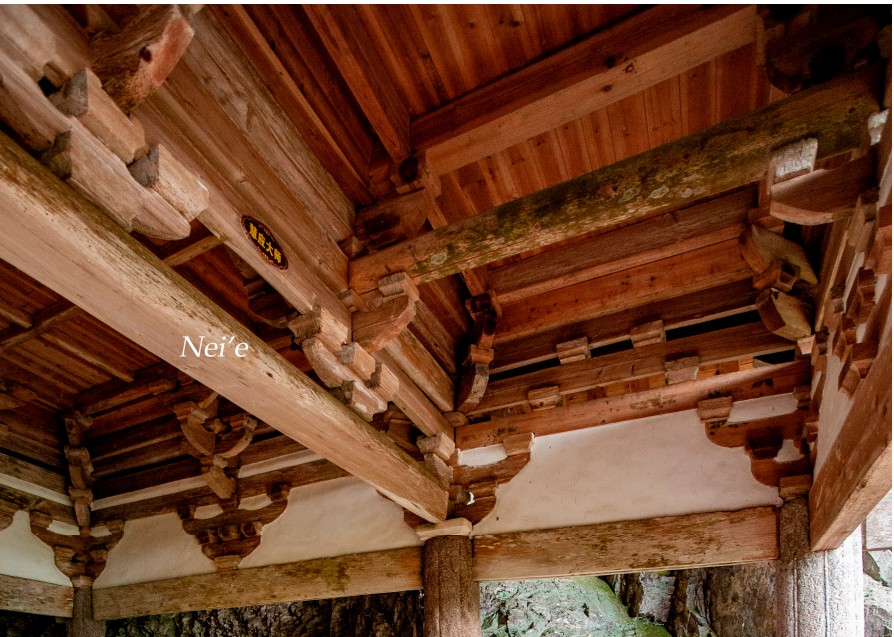

**Figure 7.** Internal longitudinal architrave connected with bracket sets to bear the beams in the Zushi Hall of Mingshan Temple in Yongtai. Photography by Xiaobin Li.

This approach is a crucial technical aspect in the reconstruction of a one-bay Buddhist building, addressing the challenges of beam intersection and support without internal columns. The internal architrave not only needs to be as long as the entire width of the structure but also needs to bear the loads from the central overhead bracket sets, beams, and the rough structural frame. Consequently, its cross-section must be large enough.

While there are no early architectural remains reflecting this technique in the Changxi River Basin or even in Fujian, similar cases of using massive internal longitudinal architraves to support upper structures can be widely found in the mansion halls of residential houses from the Ming Dynasty in eastern Fujian. Locally, internal longitudinal architraves are referred to as *kangliang* 扛梁 (Figure 8) (Ruan 2016, pp. 104–8), possibly indicating the continuity and evolution of corresponding beam-bearing techniques in the region.

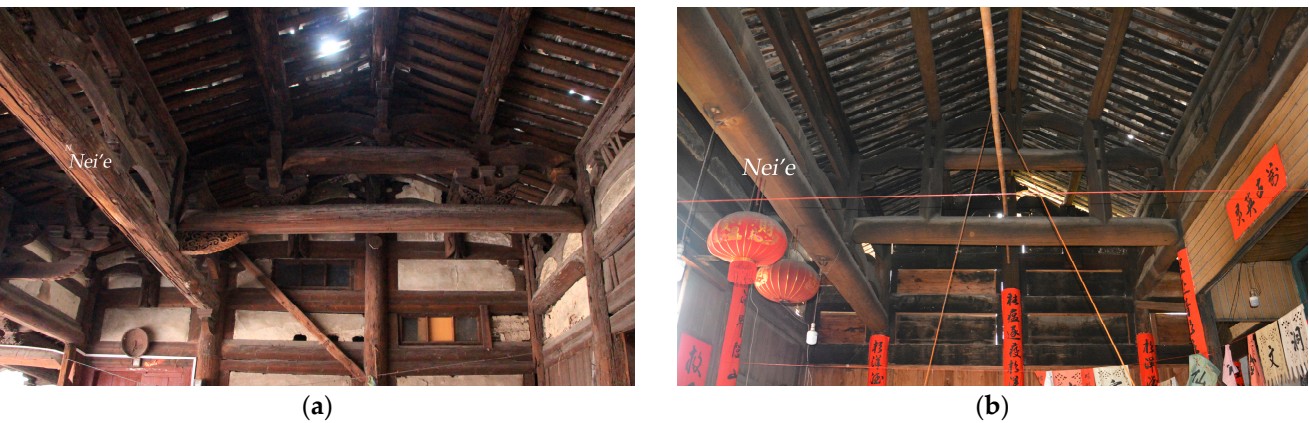

(**a**)                                                                 (**b**)

**Figure 8.** The examples of *kangliang* in residential buildings in eastern Fujian: (**a**) The Zheng's residential building in Longyuan, Cangshan district, Fuzhou city; (**b**) the hall in Qianlong, Lidun, Zhouning county, Ningde city. Photography by authors.

## 4. The Conceptual Reconstruction Plan

Combining the analysis of construction rulers, early wooden construction characteristics, and the actual conditions of the site, the main hall was reconstructed (Figures 9–13).

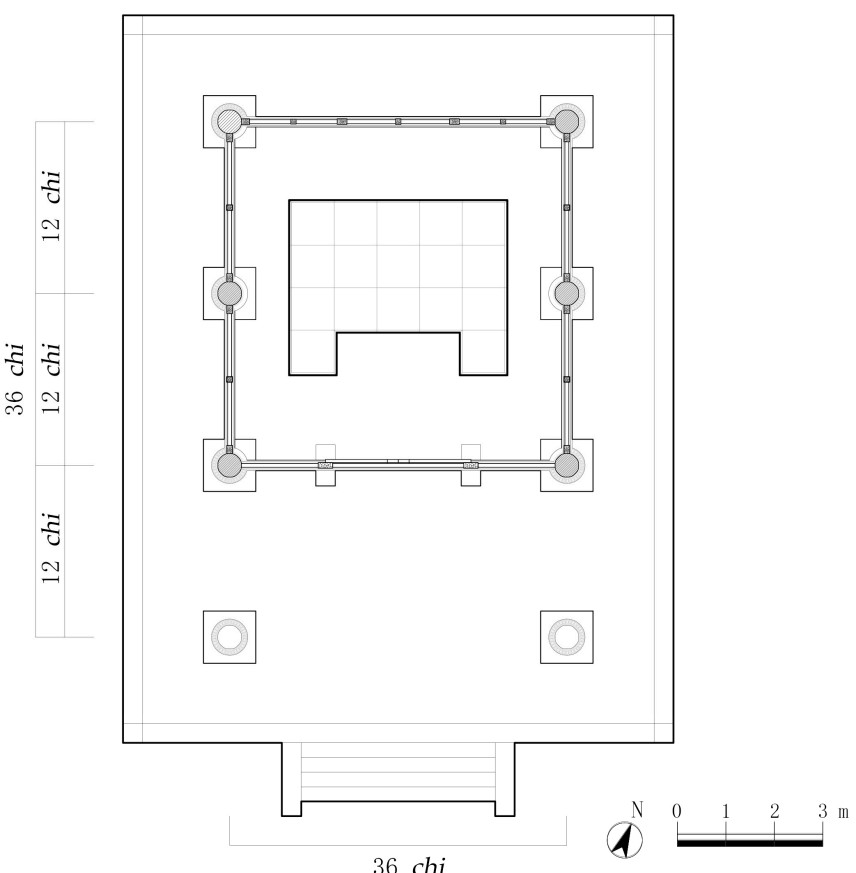

**Figure 9.** Master plan of the main hall reconstruction. Drawing by authors.

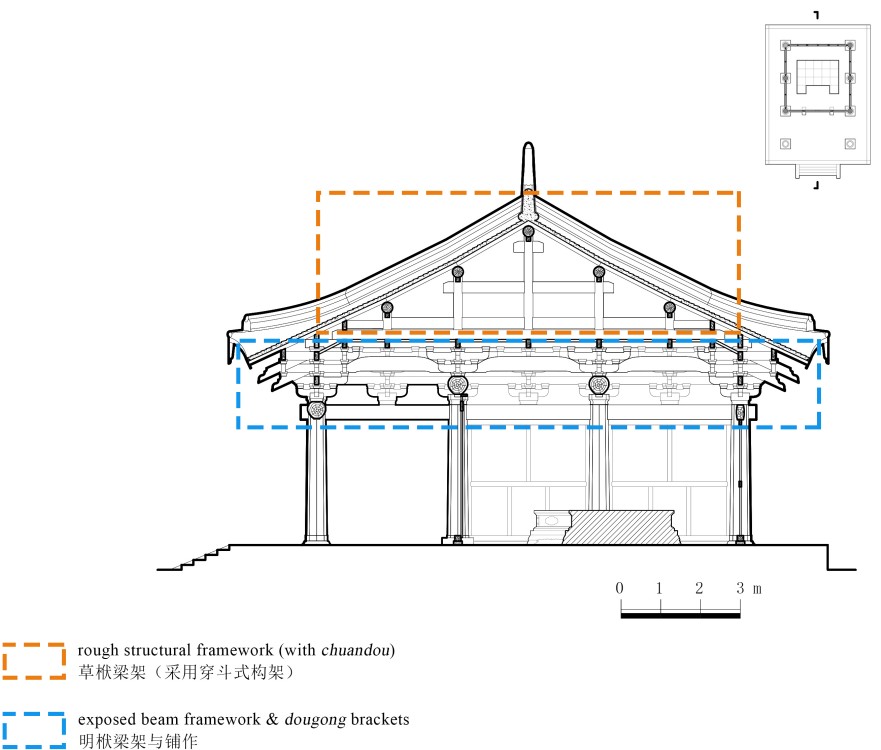

**Figure 10.** Transverse section of the main hall reconstruction. Drawing by authors.

Taking 294 mm as the basic unit of the construction ruler, the main hall has a one-bay width of 24 *chi*, a depth of three bays, each measuring 12 *chi*, a six-purlin-rafter 六架椽, with each purlin measuring 6 *chi*, and a hip-gable roof. The main hall has a palatial hall frame with a front open corridor and an enclosed rear corridor. Above the pumpkin-shaped circular stone column, the longitudinal architrave supports the bracket layer. Two inter-columnar bracket sets are located in the front and rear eaves, and each bay in the transverse direction has one intercolumnar bracket set.

The second and third rows of columns use internal architraves to support the beam. The massive, rounded architraves connect to the brackets on top of the columns, and two intercolumnar bracket sets rest above them. In the transverse direction of each bay, the rounded moon-shaped beams are used to connect the front and rear bracket sets, support the upper ceiling, and create a harmonious and dignified interior space. Above the ceiling, the rough structural framework *chuandou* is used as the roof truss. The gable beam frame employs two rafters. The tail of the upper rafter is connected at the location where the rough structural frame passes through the tie beam. The tail ends of the eave rafters on the gable side are arranged outside this beam frame to extend the position of the starting point of *chuji*, so that the bargeboard is able to be placed along the column axes at the eastern and western eaves without *shoushan*.

Original materials such as tiles and small woodwork 小木作 components from the Song Dynasty in Fujian are almost non-existent. In light of this, the reconstruction plan's imagery for the *chiwen* 鸱吻 is referenced from the precious physical objects of Southern Song found in the Shen Pavilion of the Ganlu Temple in Taining (B. Zhang 1982, p. 128). While the door and window styles tentatively adopt common Song Dynasty patterns, which are shown in *Yingzao Fashi* (Pan and He 2005, pp. 110–19).

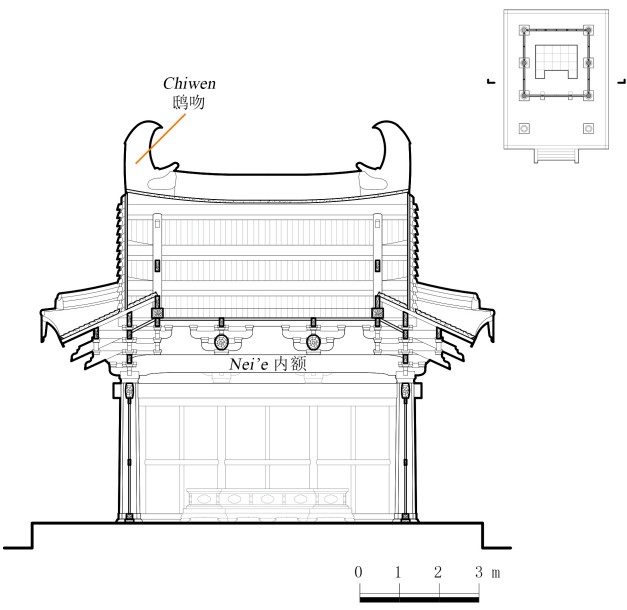

**Figure 11.** Longitudinal section of the main hall reconstruction. Drawing by authors.

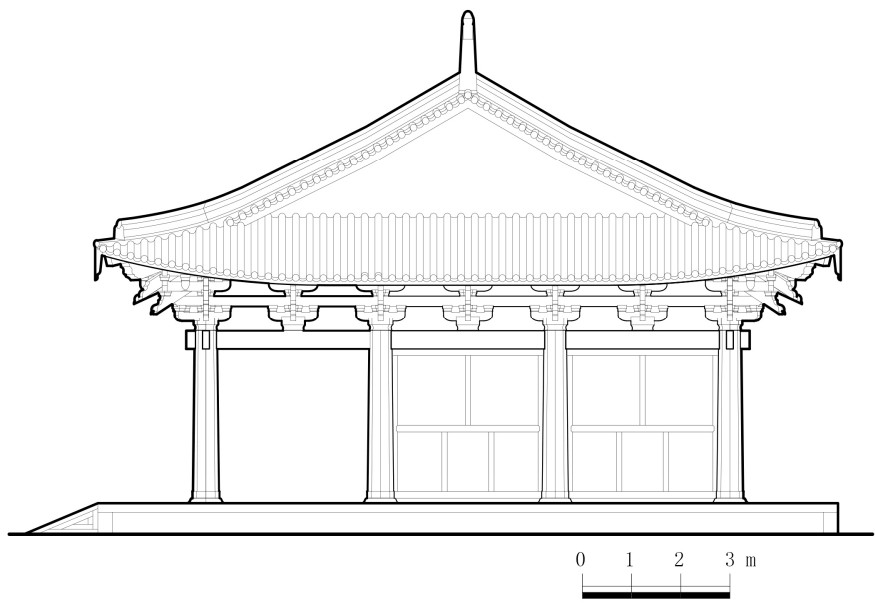

**Figure 12.** East side elevation of the main hall reconstruction. Drawing by authors.

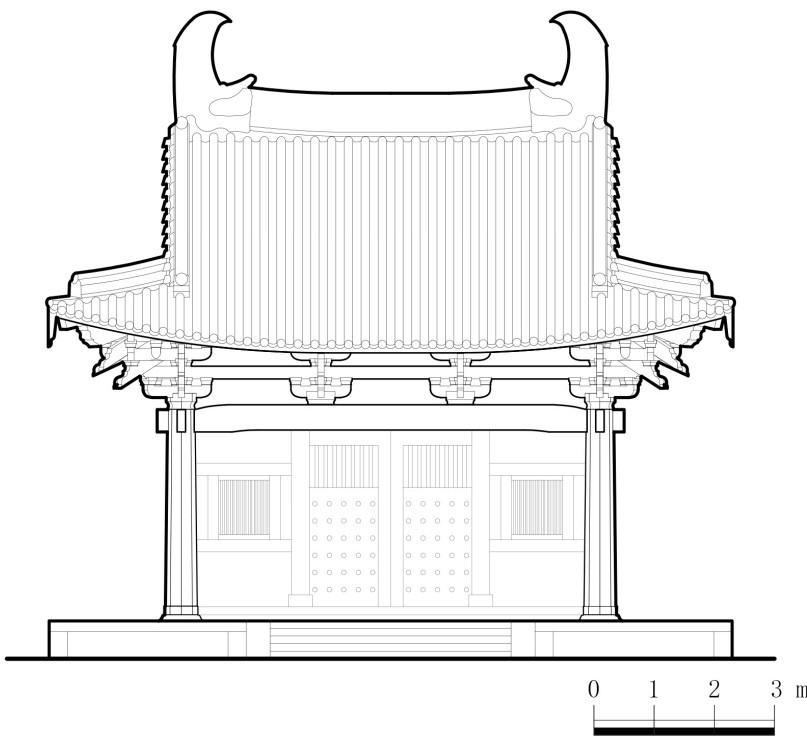

**Figure 13.** Front elevation of the main hall reconstruction. Drawing by authors.

## 5. Conclusions: The Unique Construction System in Changxi River Basin

Buddhist architecture, one bay in width and multiple bays in depth, is a unique architectural type in the Changxi River Basin of eastern Fujian, prevalent during the Tang and Song dynasties and limited to a relatively enclosed geographic unit. Using the site of the Gonghoulou Temple in Huotong Town as the subject of the reconstruction study and through an examination of early surviving architecture and regionally significant examples, a distinctive construction system can be derived for the Changxi River Basin, one that encompasses a layout of the ground plan, modular units of the construction ruler utilized, and timber frame construction techniques.

This Buddhist architecture adopts a longitudinal rectangular ground plan, one-bay in width and three bays in depth. Based on the form of the remaining stone columns, carvings at their bases indicate the spatial enclosure of the main hall. Specifically, the first bay is an open front corridor, while the subsequent two bays form a closed square Buddhist space, aligning with the common spatial arrangement of Buddhist halls in the Tang and Song Dynasties (Steinhardt 2022, p. 183). Considering the scale of the concave-shaped Sumeru podium, the main hall had few Buddha statues, likely featuring only the principal Buddha with attendant bodhisattvas positioned on two sides, and no additional statues surrounding the space, creating a small and appropriate spatial arrangement. The use of a square Buddhist space with a concave-shaped Sumeru pedestal was common in early periods, as seen in Cave 205 (Early Tang), Cave 196 (Late Tang), and Cave 55 (Northern Song) in the Mogao Grottoes in Dunhuang, as well as the main hall of the Nanchan Temple in Wutai Mountain, Shanxi (Tang), and the Yuhua Palace in Yongshou Temple, Yuci (Northern Song) (Huang 2013, p. 57). Scholars have pointed out that the reason for early Buddhist architecture providing only the principal Buddha as opposed to later periods is related to the development of Buddhist Pure Land beliefs (Ding et al. 2021). During the Five Dynasties and Northern Song period in the Jiangnan and Southeast regions, statues of Arhats were often independently placed alongside other Buddha sculptures separate from the principal Buddha. However, by the Ming and Qing periods, with the widespread prosperity of Pure Land Buddhism, the Arhat statues were placed around or on both sides of the principal Buddha. In this period, the addition of eaves in small Buddhist architecture in the

Jiangnan and Southeast regions can be seen as a response to this development (Zuo 2019, pp. 119–30). Therefore, although Zen Buddhism was prevalent in this region during the Song Dynasty, which might have influenced the stylistic features of the Buddhist temples, elements such as the Sumeru stone podium, lotus-shaped column bases, and open front corridor suggest proximity to the practices of Pure Land Buddhism. After all, Ningbo's Baoguo Temple, which shares common spatial characteristics, serves as evidence. Its affiliation with the Tiantai Sect also incorporates the beliefs of Pure Land Buddhism. The Pure Land pool in front of the Baoguo Temple main hall was also constructed in the early Southern Song Dynasty (Ding et al. 2021).

The construction ruler used for this Buddhist architecture was determined by the study to be 294 mm, slightly smaller than the official Northern Song ruler of 310 mm. However, it aligns with the tradition in eastern Fujian of using local rulers. The Southern Song's *San Shan Zhi* describes the situation during the Wuyue Kingdom, where one local ruler was equivalent to eight *chi* and seven *cun* 寸 of the official ruler, when they built the road around West Lake in Fuzhou. (Today it is estimated that a single *chi* of this local ruler equals 270–274 mm.) The length of the Southern Song wooden ruler excavated from Huang Sheng's tomb in Fuzhou, Fujian Provincial Museum, is 283 mm. Scholars have also reconstructed the construction ruler of the main hall of Hualin Temple, determining it to be 289 mm, showing the possibility of a distinct and smaller local ruler in eastern Fujian (Sun 2012, pp. 71–72). The main hall was reconstructed based on a width of 24 *chi* and a depth of 36 *chi*, with each bay measuring 12 *chi*. The ratio of width to depth is 2:3, closely matching the proportions found in the Song Dynasty Buddhist architecture site of Guoxing Temple at the Taimu mountain in Xiapu County. The column system at the site is also one bay in width and three bays in depth, with a restored construction ruler determined to be 292 mm per 1 *chi* (Appendix A). The question of whether this is representative of a more widespread relationship during this period should be further studied.

In terms of the timber frame construction of the main hall, it is different from the square, three-bay lattice-shaped structure in the Jiangnan region (S. Zhang 2015). In the Changxi River Basin in eastern Fujian, the frame construction of the one-bay palatial hall with a large dimension and hip-gable roof truss presented significant challenges in the area of timber construction. The use of internal architrave for bearing beams in the hall is crucial for transferring the upper beam loads and forming the hip-gable roof truss under the condition of no interior columns. This reflects an ancient logic of longitudinal framing. While the proposed reconstruction plan is subjective, it is likely the most suitable choice for one-bay Buddhist architecture because of the comparison with related timber frame construction and the use of the *kangliang* in later residential buildings in eastern Fujian.

The regional characteristics of the construction system are derived from the comparative analysis with existing wooden architectural remains in Fujian, timber frame construction in the Tang and Song Dynasties, and *Yingzao Fashi*. It reflects the regional and temporal nature of one-bay Buddhist architecture in the Changxi River Basin during the Song Dynasty. These features continued to evolve and be inherited in local Buddhist architecture during the Ming and Qing periods. The reconstruction of the Gonghoulou Temple provides a valuable sample of early Buddhist architecture in the Changxi River Basin in eastern Fujian, offering important insight into the exploration of timber frame construction technology and its historical development in Fujian.

**Author Contributions:** Conceptualization, J.L.; methodology, J.L. and Y.D.; software, Y.D.; formal analysis, J.L. and Y.D.; investigation, J.L. and Y.D.; resources, J.L. and Y.D.; writing—original draft preparation, Y.D. and Y.C.; writing—review and editing, J.L., Y.D. and Y.C.; visualization, Y.D. and Y.C.; supervision, J.L.; project administration, J.L.; funding acquisition, J.L. All authors have read and agreed to the published version of the manuscript.

**Funding:** This research was funded by [National Natural Science Foundation of China] grant number [50308015].

**Institutional Review Board Statement:** Not applicable.

**Informed Consent Statement:** Not applicable.

**Data Availability Statement:** Data are contained within the article.

**Conflicts of Interest:** The authors declare no conflicts of interest.

**Appendix A**

**Table A1.** The Restored Construction Ruler of the Main Hall of Guoxing Temple in Taimu Mountain (1 *chi* = 292 mm).

|  | Total Width | The First Bay of the Depth | The Second Bay of the Depth | The Third Bay of the Depth | Total Depth |
|---|---|---|---|---|---|
| mm | 7051 | 3504 | 3506 | 3509 | 10,519 |
| *chi* | 24.147 | 12.0 | 12.006 | 12.017 | 36.024 |
| Round up | 24 | 12 | 12 | 12 | 36 |
| Rate of Coincidence | 99.39% | 100% | 99.94% | 99.86% | 99.93% |

Data source: Measured on site by authors.

**Notes**

[1]  In ancient Chinese architecture, the width of bays generally adheres to the principle of integer lengths between columns, which means that the width of the central bay and secondary bays is based on integer *chi*, half *chi*, and occasionally 1/4 *chi*, allowing for clear dimensions of architectural components, which facilitates estimation, design, and construction. This principle can be observed in architectures from the Tang Dynasty to the Qing Dynasty. The principle serves as the theoretical basis for the reconstruction of construction rulers.

[2]  The height-to-thickness ratio of *cai* is 2:1 in the early wooden structures in Fujian, thus it raises the question of how to correspond the *cai* module according to *Yingzao Fashi*. Specifically, it is necessary to first calculate the *fen* value and then to respectively correspond dimensions of *cai*, i.e., 15 *fen* × 7.5 *fen* or 20 *fen* × *cai* thickness of 10 *fen*.

[3]  Modern architectural historians have named the two main types of wooden frame structures in China as "*tailiang*" 抬梁 and "*chuandou*" 穿斗, which are modern terms. While "*diange*" and "*tingtang*" are historical terms defined in the Song Dynasty's *Yingzao Fashi*, and they classify one type of *tailiang* structure. The corresponding frame structures can be called *diange*-style structure and *tingtang*-style structure.

[4]  "Chuji" and "Shoushan" are two common construction methods for the gable on hip roof. "Chuji" refers to protruding beyond the edge of a gable wall. "Shoushan" involves pulling the bargeboard inward by a certain distance, preventing the roof from becoming overly massive.

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
