# Peer review of "Reconstruction of Single-Bay Buddhist Architecture Based on Stylistic Comparisons in Northeast Fujian, the Core Hinterland of the Changxi River Basin—Using Gonghoulong Temple as an Example"

_religions, doi:10.3390/rel15040474_

Round 1

Reviewer 1 Report

Comments and Suggestions for Authors

This article is focused on reconstructing what appears to be a particular type of Buddhist architecture found in Fujian province. The article is described as being a companion piece to an earlier article regarding the archaeological evidence for this type, of which there are seven examples. The type consists of a single bay in width, and three bays in depth, which is effectively shown to be a ritual hall with a single bay portico and a 1 x 2 bay enclosed interior. The varied decoration of the column bases was particularly convincing in this regard. Overall, I found the article to be convincing in describing this as a particular type reflecting a regional style of structure specific to Fujian but related to other buildings of the Song-Yuan period in Southeast China. This is an important discovery, and the detailed technical discussion is warranted. Additionally, it would be helpful to know more about the ritual significance of this configuration through comparisons with extant examples.

The technical discussion, while very important to the argument, is not sufficiently clear to the reader. This is partly due to the translation from Chinese into English and partly due to the difficulty of the technical architecture terminology more broadly. Below are some suggestions which would help make the reconstruction more understandable to readers with a background in Chinese architectural history as well as Chinese religions.

1.     Authors should cite the Yingzao fashi directly when discussing the contents of that text. I did not see this work in the bibliography. Furthermore, citations to the Yingzao fashi should be given by chapter (juan ) number and page number from the edition being used.

2.     Proportions: The modular system of Chinese architecture may not be fully understood to the reader of Religions. Thus, when discussing measurement and proportion it would help to explain why you are concerned with converting the measurements into chi and what the significance of the cai and zhi are. This should be defined and illustrated.

3.     On page 8, proportions are discussed in ratio form (e.g. 3:2) in the text, but are described in decimal form (e.g. 1.5) in the table. This is confusing.  Perhaps both could be used in both text and table?

4.     The discussion of choice of diange vs tingtang, and the relationship of these Song-period terms to modern structural terms such as chuandou (p. 9) is not clear. These terms are not familiar to most English-speaking scholars outside of Chinese architecture history. They are valuable for understand ritual architecture, however, and are therefore important to clarify. More explanation would be very helpful. For example, how does one determine whether it is reasonable to use diantang or tingtang for a building that doesn’t have interior columns? On p. 7 and p. 9 the authors discuss similarities to the main hall of Baoguo Temple in Zhejiang. Is the main hall of Baoguo Temple tingtang or diantang? What about Hualin Monastery main hall? Or, perhaps more importantly what about the Nanchan Monastery Main Hall, which does not have any interior columns? This is not sufficiently explained. Additionally, a summary definition of “diangeshi goujia should be given, ideally with a citation to the source for this definition. Examples of why the building should *not* be tingtang are as important as why it *should* be diantang. When used in the text, other technical terms should also appear on drawings so that the reader can follow the argument presented in the text.

Comments on the Quality of English Language

1.     The English on this article is good overall. However, the use of technical language could be improved for clarity. For example, “relics” in a Buddhist context refers to the relics of the Buddha (Ch. sheli 舍利). Here it is being used to discuss archaeological remains. This is confusing.

2.     The terms used to translate the architectural vocabulary are frequently incorrect. For example, the term “basin stone” (l. 156) is not the English term for plinth or pillar base. Also when discussing an individual building, use “building” not “architecture” (as seen in the title). These are only two examples. All of the technical vocabulary should be scrutinized by a native English speaker who is familiar with specialized architectural vocabulary. I recommend using the Getty Art and Architectural Thesaurus (or similar) to help identify the correct English architectural term.

Author Response

Dear Reviewer:

Greetings. Thank you for the invaluable feedback, which has greatly contributed to the further refinement of this manuscript. Your suggestions have prompted us to delve deeper into areas needing improvement, thereby enhancing the readability of the article. The following is a brief overview of the modifications made as a response to the feedback:

1. References:

We have added citations from Yingzao Fashi and properly referenced the volume numbers and page numbers in the text. Additional international literature and previous studies on Yingzao Fashi have been included.

2.Improvements for Terminology and aiding foreign scholars in architecture and religious studies:

Detailed annotations and explanations have been provided for various concepts, such as:

the "cai-fen system 材份制", “cai 材” and ”zhi 栔” (p. 8) within the Chinese modular system;

the "principle of integer lengths between columns 整数尺柱间制" (Note 1);

the historical terms “diange-style structure 殿阁式构架” and “tingtang-style structure 厅堂式构架” in Yingzao Fashi as well as the modern terms “tailiang structure 抬梁式构架” and “chuandou structure 穿斗式构架”, including their connotations and interrelationships (Note 3).

In particular, the definitions and distinctions between the diange-style structure and the tingtang-style structure are cited in detail, with Zhang Shixing's three indicators for distinguishing between them (p.10), to help readers in their understanding.

3. Numeric expressions on page 8 have been reformatted into proportional form (e.g., changing 1.85 to 1.85:1).

4.Addressing academic concerns raised by the reviewers:

A detailed response to the question of why the interior hall space without the columns should not be tingtang-style structure is given at p.11.

Here, I would like to add a supplement. In fact, there is a wide variety of existing cases of traditional wooden structures, particularly in the southern regions, which significantly differ from the original context of "diange" and "tingtang" in Yingzao Fashi. Futhurmore, the architectural column grid studied in this manuscript does not align with the several "diange" patterns documented in Yingzao Fashi. Because there is no mention of column grids with single bays and multiple depths in Yingzao Fashi. However, the horizontal stratification logic of the structure is still observed. Therefore, the choice of framework type we proposed is based on logical inference, indicating a stronger likelihood of it being a "diange-style structure". While the possibility of it being a "tingtang-style structure" exists, it is deemed less probable. A detailed analysis of this matter is provided on page 11.

Nonetheless, the question raised by the reviewer is highly thought-provoking and has aided in further contemplation. We deeply appreciate it!

Additionally, the figures have been revised with increased annotations of important technical terminology within the images.

5. Efforts have been made to standardize and unify technical terminology.

For example, "relics" has been replaced with "remains", and "column base" is consistently used for "柱础". Although the term "protruding basin stone" lacks a corresponding English word, we have opted to retain it. We have consulted the Getty Art and Architectural Thesaurus and Architectura Sinica organized by Southeast University, China (https://architecturasinica.org/) for relevant technical terminology. 

To enhance the English expression, assistance from an American doctoral student specializing in the history of ancient Chinese architecture at Harvard University has been sought for further refinement.

Reviewer 2 Report

Comments and Suggestions for Authors

This article is interesting, well written and well-illustrated, which brings to light the reconstruction of a regional style of architecture that disappeared in the course of history. Its contribution is significant in recovering unique characteristics of a type of Buddhist architecture belonging to the Tang and Song dynasties through the analysis of on-site evidence, historical records, and stylistic comparisons. 

It would be interesting to see a bibliography that includes the sources of more international authors. 

The need to support the analysis in the detailed description of the architectural elements for the reconstruction is understood, but since it is a journal about religion, it is advisable to also align it with the interest in the religious-cultural aspect. 

It is not mentioned to which branch of Buddhism this temple belonged. But due to the popularity of the Chan school in the area, some construction aspects may have been due to stylistic forms typical of this branch. On the other hand, elements such as the Sumeru stone podium, the columns carved with lotuses, and the open front corridor suggest a closeness to Pure Land Buddhist practice. 

The reconstruction shown on page 16 (figure 13 – front elevation) reveals a building with a hip-gable roof (Xieshan) typical of the Tang and Song dynasties. The roof ridge draws attention because the reason for its exterior design is not mentioned, and whether it was taken from the examples shown in Yingzao Fashi’s architectural treatise.

For a better and easier reading of the article, it is recommended to make the following changes:

Page 3 (line 79): open a space between the ideogram and the parenthesis.

Page 3 (line 86): open a space between the ideogram and the parenthesis.

Page 3 (line 87): open a space between the ideogram and the parenthesis.

Page 3 (line 88): open a space between the ideogram and the parenthesis.

Page 4 (line 101): (J. Liu 2018) does not appear in the references.

Page 4 (line 104): create a space above to separate the subtitle from the previous paragraph. Also, put the subtitle in bold.

Page 16 (line 435): move the written content below figure 11 on page 15.

Page 17 (line 486): put the e. (Th à The).

Author Response

Dear Reviewer:

Greetings. Thank you for the invaluable feedback, which has greatly contributed to the further refinement of this manuscript. Your suggestions have prompted us to delve deeper into areas needing improvement, thereby enhancing the readability of the article. The following is a brief overview of the modifications made as a response to the feedback:

1.References:

We have added citations from Yingzao Fashi and properly referenced the volume numbers and page numbers in the text. Additional international literature and previous studies on Yingzao Fashi have been included. We have expanded the list of English references, such as the literature of Guo Qunghua and Feng Jiren that have greater relevance to this paper. Furthermore, an article by Steinhardt on the broader study of Chinese ancient architecture and an article by Zuo lala regarding Yuan Dynasty architecture in southern China are also included.

Due to the fact that there are very few international publications with direct relevance to this research thesis, we have made effort to include relevant citations. Any further recommendations regarding significant international scholars' research would be highly appreciated.

2.Improvements for Terminology and aiding foreign scholars in architecture and religious studies:

First of all, detailed annotations and explanations have been provided for various concepts, such as:

the "cai-fen system 材份制", “cai 材” and ”zhi 栔” (p. 8) within the Chinese modular system;

the "principle of integer lengths between columns 整数尺柱间制" (Note 1);

the historical terms “diange-style structure 殿阁式构架” and “tingtang-style structure 厅堂式构架” in Yingzao Fashi as well as the modern terms “tailiang structure 抬梁式构架” and “chuandou structure 穿斗式构架”, including their connotations and interrelationships (Note 3).

In particular, the definitions and distinctions between the diange-style structure and the tingtang-style structure are cited in detail, with Zhang Shixing's three indicators for distinguishing between them (p.10), to help readers in their understanding.

Additionally, the figures have been revised with increased annotations of important technical terminology within the images.

Finally, the manuscript has been updated to include a response to and speculation on the Buddhist sect to which the temple belongs (p.19), aiming to enhance the readability of the paper for scholars of religious studies.

3.Addressing academic concerns raised by the reviewers:

A response has been provided on page 16 regarding the basis for the depiction of chiwen 鸱吻 features in the restoration drawings, addressing the concern raised by the reviewer.

4.Modifications have been made to address the formatting issues pointed out by the reviewers.

The above-mentioned changes have been highlighted in Word document. Once again, we express our gratitude to the reviewers for the valuable feedback on the revisions made to this manuscript and eagerly await your response.